# Examining the Effectiveness of the Educational Program Developed for English Teachers Working with Students Aged 13–18 Who Have Specific Learning Disability

**DOI:** 10.3390/children10010081

**Published:** 2022-12-30

**Authors:** Yaşar Uçak, Mukaddes Sakalli Demirok

**Affiliations:** Special Education Department, Atatürk Education Faculty, Near East University, 99138 Nicosia, Cyprus

**Keywords:** specific learning disability, curriculum, English language teachers

## Abstract

The current study examined the competencies of English teachers working with students having a special learning disability (SLD) and the effectiveness of the developed education program. The study was made during the pandemic, and the research group consists of voluntary 28 English teachers working at the level of second and third schools in the districts of Konya in the 2021–2022 academic year. A mixed method (qualitative + quantitative) was used in the research. The research was designed with a pretest-posttest control group experimental design. In the study, the proficiency levels of the teachers who participated in the training program in the sub-dimensions of SLD knowledge, SLD academic skills, SLD professional knowledge, and foreign language teaching knowledge were found to be statistically significantly higher than both the pre-test scores and the teachers who did not participate in the training. In addition, it was determined that there was a statistically significant and positive correlation between all sub-dimensions of English teachers’ proficiency levels in the field of SLD. These results indicate that the training program on SLD plays an important role in increasing the competence of English teachers in this field. Evidence obtained in the study will guide experts who prepare teacher training and in-service training programs.

## 1. Introduction

Learning is a lifelong mental activity that starts with the birth of the person and acquires different skills and abilities depending on the developmental stages. In this learning process, school periods have a prominent place for the individual’s self-development and psychology. Because school periods are the first step in which individuals acquire academic equipment and compare themselves with their peers.

Along with school experience, students are expected to be able to fulfill various duties and responsibilities. It is predicted that learning difficulties, which are understood when the student cannot exhibit the same skills as their peers in academic fields such as reading, writing, and mathematics, affect the individual’s developing self-perception, mental health, and peer acceptance [1]. At this point, the first step is to be taken in the process of diagnosing students with special learning difficulties and determining the characteristics of their difficulties.

As important as the diagnosis of SLD, the prevalence of these children is increasing day by day, which is the largest group in special education and is seen to cover approximately 50% [2] is that the education to be given to these students is not enough. planning and taking necessary action. The strengths and weaknesses of these students, who should receive education both individually and in groups, should be determined and an education program should be organized accordingly [2]. At this point, in terms of the operability of the education to be given, the sufficient knowledge and skills of the educators who work with the students with SLD in schools, that is, the teacher competencies play a major role. Because “preparing suitable environments by determining learning objectives for students with different special education needs” is considered a performance indicator for a teacher’s professional development competence [3]. This situation should be seen as an important reference to the steps that teachers will take to develop their competencies and gain new competencies in the age of information and technology we live in.

The main purpose of this study; The aim of this study is to examine the effectiveness of the ‘teacher competencies training program’ developed for English teachers working with adolescents aged 13–18 with special learning disabilities. In this direction, English teachers working with adolescent students between the ages of 13–18 with special learning difficulties have “definition, diagnosis and evaluation, causes, classification, developmental characteristics, instructional planning and adaptation, legal rights, family education and guidance, special education” about these students with SLD. methods and techniques, teaching technology and material development, coping with problem behaviors in the classroom, measurement and evaluation, IEP development and application, and education in Turkey” were investigated.

When the domestic and foreign literature on the subject is examined, it is seen that the subject of special learning difficulties has increasing popularity in recent years [4] and studies are generally conducted on a student sample [5,6,7]. Studies with teachers as the research group generally proceed with qualitative findings based on opinions [4,8,9,10,11], educational programs It has been observed that the studies in which teacher competencies are tried to be determined and measured are quite limited [12,13,14,15]. In fact, it has been determined that no research has been conducted on the determination of the proficiency of foreign language teachers in teaching English to students with SLD. This research is important both in terms of determining the competence of English teachers in the field of SLD and determining the effectiveness of a newly developed education program on SLD. In this context, it is predicted that the added value that our research findings will provide to the field is quite high.

In line with this stated purpose, answers to the following questions were sought in the study:Sub-problem: Is there a difference in the proficiency levels of the teachers in the experimental and control groups (content knowledge, professional knowledge, academic skills teaching knowledge, foreign language teaching knowledge), compared to before and after the applied education program?Sub-problem: Is there a significant difference in the proficiency levels of the teachers in the experimental group (content knowledge, professional knowledge, academic skills teaching knowledge, foreign language teaching knowledge), according to the pre-test and post-tests?Sub-problem: Is there a significant correlation between the proficiency levels of SLD (field knowledge, professional knowledge, academic skills teaching knowledge, foreign language teaching knowledge) of the teachers in the experimental group?

## 2. Conceptual Framework

In this section, a theoretical framework related to the subject of the research has been created.

### 2.1. The Concept of Special Learning Disability

The concept of learning disability, which is used to describe children with academic retardation despite the absence of mental retardation, was first used in 1963. However, the content of the concept is still considered one of the confusing issues since then. In many fields such as medicine, psychology, and educational sciences, discussions continue among experts and researchers about the definition of the concept of learning disability [16].

Specific learning disability (SLD) is specifically defined as “individuals with normal or above-normal intelligence (IQ > 85) who do not have any obvious brain pathology or sensory impairment, but who have severe difficulties in listening, speaking, reading, writing, reasoning and understanding, and mathematical persons who have severe difficulties in acquiring and using skills; Secondly, it is defined as the situation in people who have problems in self-management, social perception and communication, and who cannot perform in accordance with their age and intelligence despite the standard education [17]. The British Dyslexia Association has defined SLD as “a complex neurological picture in which reading, spelling and written language domains and learning functions are affected. It has been stated that one or more of these children’s number, note, motor function, and organizational skills are affected [18]. It is reported that these difficulties experienced by children in acquiring and using the skills of reading, writing, mathematics or listening, speaking, and reasoning are also reflected in their difficulties in self-control and social skills, along with a problem related to information processing processes and learning ability [19].

Special learning disability, which develops due to hereditary, functional, or environmental reasons, usually begins to show its characteristics more when the child meets school. Especially when the child, who starts to encounter academic skills, starts to fall behind his peers, he is noticed by his teachers and family [20]. Since it is difficult to diagnose LD, experts working on this subject should conduct a meticulous and careful study. A detailed examination in the fields of psychiatry, medicine, and education is required in the diagnosis process [21]. Learning difficulties can be seen in different ways in each child with SLD, and different academic gaps may emerge. For this reason, each child with a learning disability needs to be evaluated individually. For example; While a student with a learning disability may have problems in reading and language, another student may have problems in mathematics. In this context, it is necessary to classify the characteristics of each child with special learning difficulties and to categorize the learning difficulties experienced [22].

In 1989, SLD was grouped under two headings: developmental and academic. While academic difficulties generally cover subjects such as reading, writing, and arithmetic encountered in the educational environment, developmental difficulties include language, motor skills, balance, motor coordination, and thinking [23,24,25]. Dyslexia is the first and most widely known reading disorder that comes to mind when SLD is mentioned. Since the majority of children with SLD are children with dyslexia, the concept of reading disorder/dyslexia is considered synonymous with the concept of special learning disability in the literature [26]. However, there are other types of SLD. While the Ministry of National Education [2] treats learning disabilities in three groups as dyslexia, dysgraphia, and dyscalculia, experts approach the classification of SLD more broadly and say “reading disorder (dyslexia), written expression disorder (dysgraphia), math disorder (dyscalculia), difficulty in motor planning. (Dyspraxia), auditory processing disorder, visual perception/visual motor disability, and language disorder [27]. These mentioned disorders and inadequacies cause students to experience some important problems in both education and social life compared to their peers. These areas are generally; They are problems experienced in reading skills, arithmetic skills, language, and speaking skills, writing skills, handwriting, spelling, written expression, and psycho-social areas [20,28,29,30,31,32,33].

### 2.2. Foreign Language Teaching to Students with Special Learning Disabilities

Some important issues should be considered in the education of these students who do not have significant differences from their peers physically and mentally, but who need special education. Because the differences between students who have special learning difficulties and need special education also affect their foreign language learning. Because students with SLD probably experience the difficulties they experience in their native language while reading, writing, and trying to make sense of what they read, even more, when learning a different language. In the related literature, the studies on the learning of a language other than their mother tongue by students with special needs in education point to important findings.

Geneese, Paradis, and Crago [34] report that it is an important precaution protocol to distinguish between normal individuals who learn the English language in addition to their mother tongue and students who have problems with language development. Lundberg [35] in his research on foreign language learning and writing, reading, and comprehension of students with dyslexia was based on the discipline of cognitive psychology and pointed out the necessity of developing phonological awareness in these students during education. Baca and Cervantes [36] examined how students with certain learning disabilities can make the most of a foreign language teaching program.

He emphasized that it is imperative to develop cognitive and language skills with examples from real lives of individuals in bilingual programs in special education. In addition, at this point, the language proficiency of the student is brought to the fore; In line with lifelong learning strategies, it was stated that short-term, concrete, achievable goals should be selected with an appropriate planning.

Schwarz [37] investigated the difficulties faced by students with SLD while learning a language; He claimed that failure to learn a foreign language and low performance stemmed from avoiding making mistakes, not showing the necessary effort and motivation, and lack of experience or skill in language learning. Schwarz [37] stated that there are two difficulties in providing foreign language education in schools to students with SLD; “Educational institutions rarely reserve a section or class for these students, and it is difficult to find teachers trained to teach foreign languages to these students”. Topbaş [38], on the other hand, referred to the importance of desire and effort in teachers working with students with learning disabilities. The researcher stated that teachers’ having detailed information about the characteristics of their students will enable them to take additional measures that will contribute to the individual education programs, they will prepare.

These studies in the literature reveal that the training, competencies, and knowledge of teachers working with students with SLD during foreign language learning should be at a level to meet the needs of children.

## 3. Method

In this section, information about the model, universe, and sample of the research, the procedures used to collect data, the tools used to collect data, and the analysis of the data is presented.

### 3.1. Model of the Research

This research was designed with an experimental model in terms of revealing the effects of independent variables (education program on SLD) on dependent variables (teacher SLD domain proficiency). Since it was aimed to examine the effectiveness of the training program developed for English teachers working with adolescent students with special learning disabilities, the pretest-posttest control group design was preferred in this study. Experimental models are among the research models in which definitive evidence is obtained from scientific research models. Because in the experimental research model, the researcher applies comparable procedures and then aims to reach definitive evidence by examining the effects of these processes [39,40].

In Table 1, the SLD Field Teacher Efficacy Scale (T) is applied as a pre-test and post-test to the experimental and control groups and the training program (X) given to the experimental group is shown symbolically.

### 3.2. Research Population and Sample

The population of the research consists of voluntary English teachers who give education to secondary school students who need special education and have learning difficulties in the central districts of Konya. The sample of the research was stratified sampling method, in the 2021–2022 academic year, in the districts of Karatay, Meram, and Selçuklu II. and III. It has been determined using 28 English teachers working in schools at different levels.

### 3.3. Information on the Research Group

Information on the descriptive characteristics of English teachers participating in the research is shown in Table 2.

When the descriptive characteristics of the participants given in Table 2 are examined; It has been determined that 50% of the teachers are female and 50% are male, all of the teachers work in institutions affiliated with the Ministry of National Education, the average working time of the teachers in the experimental group is 15.8% and the teachers in the control group have an average of 16.57% years. In addition, it is seen that the average of the experimental group teachers is 5.81%, with a maximum of 23 years, and the average of the teachers in the control group, with a maximum of 24 years, of working with students with SLD.

### 3.4. Process

In this research, a series of process processes were designed, including before the training program, giving the training, and after the training program. In all these processes, the research in the relevant literature on the subjects of “special learning difficulties, teacher competencies, children with learning difficulties, foreign language teaching to children with learning difficulties” were scanned in-depth, and opinions were obtained by referring to expert knowledge. In the first stage of the research, the ‘Teacher Competencies Education Program for the Field of Special Learning Disability’ to be applied to the teachers was developed by following a series of steps. Then, the ‘Special Learning Disability Teacher Efficacy Scale’ was given its final form and the scale was piloted. The sample was divided into two groups, experimental and control, and pre-tests were applied to both groups.

A series of steps were followed while preparing the training program. In the first stage, the program objectives and contents were determined. The topics that English teachers should know about children with special learning difficulties have been examined in depth. Then, learning and teaching activities and evaluation activities were prepared. The developed training program was prepared under four modules and the training process was completed in 16 h, not exceeding eight sessions. The contents of the modules are; Concepts and general information about SLD, academic skills teaching information for students with SLD, foreign language teaching information about SLD, and professional information about SLD. Table 3 briefly summarizes the content of the training program.

While the training program, the content of which is presented in Table 3, was applied to the teachers of the experimental group, who gave education to students with SLD, for 2 weeks, the control group was not subjected to any process. In the last stage of the study, post-tests were applied to both groups after the training program.

### 3.5. Data Collection Tools and Analysis of Data

The questionnaire technique was used as a data collection tool in the research. This questionnaire consists of two parts, in the first part there are questions to determine some descriptive characteristics of English teachers, and in the second part there are questions about the Special Learning Disability Area Teacher Efficacy Scale (SÖGAÖYÖ). Special Learning Disability Sufficiency Scale: The ÖÖGAÖY scale is a scale developed to evaluate the competencies of teachers working with students who need special education and have learning disabilities. The scale was developed by Deniz and Sarı [15], and validity and reliability analyzes were performed. As a result of comprehensive literature research, a scale consisting of 40 questions was prepared first, and then some items of the scale were removed from the scale as a result of Exploratory Factor Analysis (EFA) and Confirmatory Factor Analysis (CFA) tests. As a result, the Special Learning Disability Area Teacher Efficacy Scale consisting of 33 questions and three sub-factors was obtained. The first sub-dimension of the scale consists of SLD field knowledge proficiency, the second sub-dimension, SLD academic skills teaching knowledge proficiency, and the third sub-dimension, SLD professional knowledge competence. In this study, the researcher added a new sub-dimension to the scale under the name of foreign language teaching knowledge in the field of SLD to measure the proficiency levels of English teachers working with students with learning disabilities. The answers given by the participants to the questions on the scale; It was scored on a 5-point Likert scale as 1 = Strongly Disagree, 2 = Disagree, 3 = Partly Agree, 4 = Agree, and 5 = Strongly Agree.

In the development phase of the scale; Opinions were taken from 5 expert academicians working at the Near East University Institute of Educational Sciences, both in shaping the sub-dimensions of the scale and in determining the qualitative questions directed to the participating teachers in the research, and the questionnaire was given its final form.

In the study, first of all, the Kaiser-Meyer-Olkin (KMO) test was performed to determine the validity of the SAAÖY scale. As a result of the test, the KMO coefficient of the scale was found to be 0.901 and the Bartlett test value of 3610.663 and 5% were significant (*p* < 0.05). In the literature, the KMO coefficient being higher than 0.60 and the Bartlett test being significant is important in terms of showing that the sample size is sufficient [41]. The Cronbach’s Alpha reliability (internal consistency) coefficient of the ÖÖGAÖY scale, which is used to determine the proficiency levels of English teachers working with students with Special Learning Disabilities, was found to be 0.992. In determining the sub-factor groups of the scale, the maximum variability (varimax) feature, one of the vertical rotation methods, was used as the rotation method. When the Exploratory Factor Analysis results of the scale used in the research are examined; it was seen that the common variance explanatory ratios and factor load values of the scale questions were higher than the 0.30 threshold value [42].

In this direction, it has been determined that no question should be removed from the scale and that the determined questions can be used effectively in finding answers to research problems. It was determined that the total variance explanatory eigenvalues (eigenvalue) of the determined sub-dimensions of the scale were 86.73%. This value shows that the sub-factors have sufficient explanatory power [43].

The data obtained from the questionnaire applied to the research group were analyzed with the IBM SPSS version 25 package program. Repeated Measures analysis was applied during the examination of teachers’ proficiency levels in teaching students with special learning difficulties. In this direction, Multivariate Analysis of Variance (MANOVA), which includes 2 × 2 processes (progress), was used to compare the pretest and post-test results of the groups.

## 4. Results

To determine whether the proficiency levels of the experimental group of English teachers were positively affected in line with the training program developed and given; MANOVA analysis was performed to examine whether the differences between the pretest and posttest were statistically significant.

The average scores of the proficiency levels of the participant teachers in the SLD area are presented in Table 4.

When the average scores in Table 4 are examined, it is seen that the mean score of the teachers in the experimental group for SLD field knowledge proficiency levels ($\bar{x}\;$ = 3.329 ± 1.269) is the average score of the teachers in the control group ($\bar{x}\;$ = 2.076 ± 0.733), and the average score for SLD academic skills and teaching knowledge proficiency levels. ($\bar{x}$ = 3.422 ± 1.308) from the mean score of the teachers in the control group ($\bar{x}$ = 1.917 ± 0.693), and the mean score of the teachers in the experimental group for their professional knowledge proficiency level ($\bar{x}$ = 3.535 ± 1.188) from the mean score of the teachers in the control group ($\bar{x}$ = 2.135) ± 0.806), it is seen that the mean score of the teachers in the experimental group foreign language proficiency levels in the field of SLD ($\bar{x}$ = 3.500 ± 1.561) is higher than the average score of the teachers in the control group ($\bar{x}$ = 1.964 ± 1.09).

The differentiation analyses of the SLD domain proficiency levels of the participants according to the group type and test order are presented in Table 5.

When the repeated measurement MANOVA test results in Table 5 are examined; It was determined that the participant teachers’ level of SLD knowledge proficiency, SLD academic skills, teaching knowledge proficiency levels, and SLD professional knowledge proficiency levels differed statistically in terms of both group type and test order (*p* < 0.01). It was determined that the proficiency levels of foreign language teaching knowledge in the field of SLD did not differ according to the test order (*p* > 0.05), and there were significant differences in group type and group type-test order association (*p* < 0.05).

According to these results, it can be said that there is a significant increase in all proficiency levels of foreign language teachers in the field of SLD, except for foreign language teaching knowledge, both in the experimental group compared to the control group and in the posttests compared to the pretests. As for foreign language teaching knowledge, an increase was recorded in the experimental group, but there was no difference in the post-tests.

The results of the correlation analysis carried out to examine the statistical relationships between the relevant sub-dimensions that make up the teacher efficacy levels of the SLD are shown in Table 6.

When the correlation analysis results in Table 6 are examined; It was determined that there was a statistically significant and positive correlation between all sub-dimensions of the proficiency levels of English teachers in the field of SLD (*p* < 0.001).

## 5. Conclusions and Recommendations

This study explains the effect of the special learning disability field training program on increasing the proficiency of English teachers in this field. Evidence from 28 English teachers teaching adolescent students with special educational needs and learning difficulties revealed that the education program developed and implemented in the field of SLD is an effective tool in improving teacher competencies. The results obtained in the research also reveal the need for new findings in the literature on teacher competency and training programs in the field of SLD, and the need for new studies to support and generalize these findings. In addition, the results of the research emphasize the importance of program and policy developers revising their current practices on SLD training programs. Because each step that will improve teacher competencies will contribute to the sustainability of the education of students with SLD.

### 5.1. Conclusions Regarding the First Sub-Problem

When the arithmetic mean scores of the answers given by the English teachers participating in the research to the questions in the research scale were examined, it was concluded that the experimental group was higher than the control group in all sub-dimensions. In addition, it was determined that the post-test average scores of the teachers in the experimental group in all sub-dimensions were higher than the post-test average scores of the teachers in the control group (see Table 4). As a result of the fact that the average scores of SLD field knowledge proficiency levels, SLDATES (Special Learning Disability Area Teacher Efficacy Scale) academic skills teaching knowledge proficiency levels, SLDATES professional knowledge proficiency levels, and foreign language proficiency levels in the experimental group of the teachers in the experimental group were higher than both the post-training and control groups, the result of the training program applied to the teachers The SLD field reveals that it improves the proficiency levels positively.

### 5.2. Conclusions Regarding the Second Sub-Problem

In the study, it was concluded that there was a significant difference in terms of both group type and test order in the sub-dimensions of SLD field knowledge proficiency levels, SLD academic skills teaching knowledge proficiency levels and SLD professional knowledge proficiency levels. It was determined that the teachers’ level of foreign language proficiency in the SLD field differed according to group type and group type-test order association (see Table 5). This differentiation, which is seen in all sub-dimensions in favor of the experimental group, reveals that the English teachers who participated in the training program had higher SLD competencies than those who did not. This result shows that the education program has a qualified effect on the competence of teachers in the field of SLD.

The finding that the proficiency levels of English teachers who have a training program in the field of SLD differ significantly compared to those who do not have a training program can be considered a sign of the necessity of adding lessons in the field of SLD to the training programs of all teacher candidates in universities. Although English teachers were studied in this study, with the assumption that our sample represents the whole universe; It can be emphasized that training and seminars on this subject should be given to all teachers working with or not working with students with special learning difficulties. In addition, it is suggested that the curriculum developed and implemented within the scope of the research should be evaluated for practical application in in-service training seminars in MEB-affiliated and Special Education Institutions.

### 5.3. Conclusions on the Third Sub-Problem

In the results of the correlation analysis carried out to examine the relationships between the sub-dimensions of the SLDAL scale, a significant and positive relationship was obtained between the teachers’ general knowledge levels in the field of SLD, their academic skills, their professional knowledge levels, and their foreign language teaching knowledge levels (see Table 6). This result indicates that the sub-dimensions of the scale support each other and increase together. The fact that the sub-dimensions of teacher proficiency levels in the field of SLD are positively supportive of each other reveals that it is an important issue to be considered in the process of developing education programs related to the subject.

## 6. Limitations and Impact

The results of the research showed that, considering the necessity of including teachers in the intervention programs for students with SLD, education programs in the field of SLD should be used to increase teachers’ knowledge and competencies in this field. Thus, the educational needs of student groups with special learning difficulties will be shed light on. As the results of this study were obtained only with English teachers and using certain scales, there are limitations in terms of method, time, and sampling. This evidence obtained using the experimental design can be validated by new studies using different techniques and sampling. Therefore, to support the results in the literature, it is recommended to conduct new studies with larger sample groups and teachers from different branches, using similar and/or different scales.

## Figures and Tables

**Table 1 children-10-00081-t001:** Symbolic view of the research model.

Groups	Pre-Test	Process	Post-Test
Experimental Group (G1)	S1	X	S2
Control Group (P1)	S1	-	S2

**Table 2 children-10-00081-t002:** Descriptive characteristics of the participants.

Descriptive Feature Experimental Group		Experimental Group		Control Group		Total	
		n %		n %		n %	
Gender	Female	6	42.90%	8	57.10%	14	50%
	Male	8	57.10%	6	42.90%	14	50%
Graduated Faculty	Science and Literature Education	2	33.30%	4	66.70%	6	21.40%
		11	52.40%	10	47.60%	21	75.00%
	Language History	1	100.00%	0	0.00%	1	3.60%
Graduated Department	English Language and Literature	1	7.10%	4	28.60%	5	17.80%
	English teacher	10	71.40%	10	71.40%	20	71.40%
	Linguistics	1	7.10%	0	0.00%	1	3.60%
	German Teaching	1	7.10%	0	0.00%	1	3.60%
	English Philology	1	7.10%	0	0.00%	1	3.60%
Training in SLD	yes	0	0.00%	4	28.60%	4	14.30%
	No	14	100%	10	71.40%	24	85.70%
Total		14	50%	14	50%	28	100%

**Table 3 children-10-00081-t003:** The content of the SLD teacher education program.

Module Name	Content
Module 1: Concepts and General Information Related to LLD (1st and 2nd Sessions)	Information about the training program. To raise awareness about the special learning disability area, to give concepts and general information about the subject.
Module 2: Information on Teaching Academic Skills to Students with SLD (3rd and 4th Sessions)	Giving method and technical information about the academic fields in which students with special learning difficulties have problems
3rd Module: Foreign Language Teaching Information in SLD (5th and 6th Sessions)	Giving information about strategies, approaches, methods, and techniques that can be used in foreign language teaching to students with special learning difficulties.
Module 4: Professional Information on SLD (7th and 8th Sessions)	Giving professional information about students with special learning disabilities (preparation, use, evaluation, and monitoring of IEP, communication, and guidance with families, measurement and evaluation, and cooperation with teachers). Assessment of the program

**Table 4 children-10-00081-t004:** The average scores of the participants’ proficiency levels in SLD.

SLD Field Qualification Dimension	Group	Test	N	Average. ± Ss.
SLD Field Knowledge Proficiency Levels	Experimental Group	pretest	14	2.170 ± 0.663
		posttest	14	4.489 ± 0.111
		Total	28	3.329 ± 1.269
	Control Group	Pretest	14	2.450 ± 0.879
		Posttest	14	1.703 ± 0.208
		Total	28	2.076 ± 0.733
SLD Academic Skills Teaching Knowledge Proficiency Levels	Experimental Group	Pretest	14	2.257 ± 0.656
		Posttest	14	4.607 ± 0.393
		Total	28	3.432 ± 1.308
	Control Group	Pretest	14	2.221 ± 0.852
		Posttest	14	1.614 ± 0.274
		Total	28	1.917 ± 0.693
SLD Vocational Knowledge Proficiency Levels	Experimental Group	Pretest	14	2.557 ± 0.744
		Posttest	14	4.514 ± 0.562
		Total	28	3.535 ± 1.188
	Control Group	Pretest	14	2.607 ± 0.788
		Posttest	14	1.664 ± 0.500
		Total	28	2.135 ± 0.806
SLD Field Foreign Language Teaching Knowledge Proficiency Level	Experimental Group	Pretest	14	2.392 ± 0.806
		posttest	14	4.607 ± 1.332
		Total	28	3.500 ± 1.561
	Control Group	Pretest	14	2.678 ± 1.062
		Posttest	14	1.250 ± 0.518
		Total	28	1.964 ± 1.096

**Table 5 children-10-00081-t005:** Differentiation analysis of participants’ SLD domain proficiency levels according to group type and test order.

1. SLD Field Knowledge Proficiency Levels	**The dependent variable**	**F**	** *p* **	**Eta**	**Adj. R2**
	Group Type	69.230	>0.001 **	0.571	0.782
	Test Sequence	27.233	>0.001 **	0.344	
	Test Order	103.666	>0.001 **	0.666	
2. SLD Academic Skills Teaching Knowledge Proficiency Levels	**The dependent variable**	**F**	** *p* **	**Eta**	**Adj. R2**
	Group Type	92.617	>0.001 **	0.640	0.791
	Test Sequence	30.672	>0.001 **	0.371	
	Test Order	88.299	>0.001 **	0.629	
3. SLD Profession Knowledge Proficiency Levels	**The dependent variable**	**F**	** *p* **	**Eta**	**Adj. R2**
	Group Type	62.933	>0.001 **	0.548	0.712
	Test Sequence	8.258	0.006 *	0.137	
	Test Order	67.509	>0.001 **	0.565	
4. SLD Area Foreign Language Teaching Knowledge Proficiency Level	**Dependent Variable**	**F**	** *p* **	**Eta**	**Adj. R2**
	Group Type	34.524	>0.001 **	0.399	0.600
	Test Sequence	2.559	0.139	0.042	
	Test Order	48.564	>0.001 **	0.483	

* *p* > 0.05, ** *p* > 0.001.

**Table 6 children-10-00081-t006:** Correlation analysis between sub-dimensions of participants’ proficiency in SLD.

T Sub-Dimensions	Average	1	2	3	4
1. SLD Field Knowledge Competence	5.99	--	0.952 **	0.874 **	0.854 **
2. SLD Academic Skills Teaching Knowledge Competence	4.23	--	--	0.881 **	0.823 **
3. SLD Vocational Knowledge Competence	6.03	--	--	--	0.914 **
4. SLD Field Foreign Language Teaching Knowledge	3.03	--	--	--	--

** Indicates 1% significance level.

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
