# Peer review of "Examining the Effectiveness of the Educational Program Developed for English Teachers Working with Students Aged 13–18 Who Have Specific Learning Disability"

_children, 2022, doi:10.3390/children10010081_

Round 1

Reviewer 1 Report

Thank you for the opportunity to review “Examining the Effectiveness of the Educational Program Developed for English Teachers Working with Students Aged 13-18 Who Have Specific Learning Disability”. The topic examined is a worthy topic to explore, however, I have concerns before this is ready for a broader audience.

Intro/Literature Review

Provide some clear problem statement then express the purpose of the paper earlier on. It wasn’t until page 3 that the reader knew where this paper was taking us. The logical progression of the arguments could be helped with some subheadings. Extensive editing to phrasing/grammar in the literature review and discussion is needed.

Be mindful of deficit language throughout and change to strength-based language.

Example:

Pg. 1- “Because school periods are the first step in which individuals acquire academic equipment and compare themselves with their peers.” – what do you mean by “academic equipment? Do you mean when students attend school instead of “school periods?”

Pg. 1 - “In individuals with normal or above-normal intelligence (IQ>85), primarily, those who do not have a clear brain pathology or sensory disturbance, but have severe difficulties in listening, speaking, reading, writing, reasoning, and acquiring and using mathematical skills; “ – revise this is a fragment. What was the message you were trying to convey?

Methods/Results

Research questions need to be rephrased for clarity of understanding.

More explanation over the program implemented for the intervention needed.

“The fact that the KMO coefficient is higher than 0.60 and the Bartlett test is significant in the literature indicates that the scale questions are suitable for factor analysis and the scale has sufficient validity” – what type of validity? How was the instrument developed?

There needs to be extensive clarification of how and why methods were used.

For instance, I’m confused by the purpose of the semi-structured interviews. “To determine the needs of students with disabilities” is relevant, but how did you derive what was a need? What qualitative analysis was used? How were semi-structured interviews conducted and how was that information related to the quantitative information collected later?

How did you determine that the means were higher based on groups? How did you analyze a statistically significant difference between groups? Is what is reported for t-tests or only descriptive statistics? Make sure you are reporting sufficient information to make some of your claims.

“Repeated Measures tests were conducted to avoid statistically Type-1 and Type-2 errors.” – repeated measures are not a foolproof method of avoiding Type-1 and 2 errors.

Also, a DDA could be used as a post hoc to the MANOVA and would be helpful to tell the audience where the majority of the variance is coming from. 

Do not report p-values of .000 in a manuscript. If it is smaller than .001, write it until the digit out. Using “p=0.000” not acceptable reporting. This is also true for your tables. Please look over formatting guidelines for the journal or switch to APA 7th

Limitations and Implications

There were only 28 teachers who participated, and I worry about the conclusions generated from only 28 teachers participating. Do you think that you had sufficient power to run these analyses? Did you run a power analysis beforehand? What other limitations are there of this work and what are the practical implications?

Author Response

I have made the necessary correction.

Reviewer 2 Report

The papers with the topic of the use of internet with students with special needs at the age 13-18. The abstract clearly describes the main target of the research and despite the minor errors, it is clear and understandable. The literature review is up-to-date with a minor problem on citations 26, 29 and 42 that in the final bibliography have been capitalized without any apparent reason. There are no mentions to publications in 2022 but this could be caused because the research was finished only last year. 

The DOI of the final publications have not been included and those might be desirable if not necessary. 

This paper is unusual and also very welcome in the international panorama since it relates a topic that is very rarely found in academia but very necessary as well. Althought the focus on North Cyprus limits the readership of the paper, its novelty overcomes such a problem and readers will be attracted towards the content.  The conclusions summarize and clearly indicate the benefits and drawbacks of the publication.

The tables seem adequate although they should not be split as they appear in the PDF.

All in all, it is an ellegant, adequate and most welcome pepr with the necessary quality and just afew issues to revise if any.

Author Response

I made the necessary correction on it.

Thanks a lot for your recommsdation

Reviewer 3 Report

The abstract is too big. Please resize it with the objectives, method, findings, innovation, contribution/ implication. 

Is there any theoretical support in this study? If yes, please try to relate. 

References are not updated. Please use to most recent references. 

There is no research model / conceptual model / framework. Could you represent one to make the study clear? 

Conclusion is too large and vague. Please break it down into different subpoints. Make the key contribution clear. 

Good Luck!

Author Response

I made the necessary correction.

Thank you very much for your recommendation

Round 2

Reviewer 3 Report

The revised version is OK